

# Ice Nucleating Particle Concentrations over the Eurasian-Arctic seas

Guangyu Li[1,2], André Welti[3], Iris Thurnherr[1], Ulrike Lohmann[1], and Zamin A. Kanji[1]

[1]Institute for Atmospheric and Climate Science, ETH Zurich, Switzerland
[2]Laboratory for Microwave Spatial Intelligence and Cloud Platform, Deqing Academy of Satellite Applications, China
[3]Finnish Meteorological Institute, Helsinki, Finland

**Correspondence:** Guangyu Li (lgy526462219@gmail.com) and Zamin A. Kanji (zamin.kanji@env.ethz.ch)

**Abstract.**

Ice nucleating particles (INPs) catalyze primary ice formation in Arctic low-level mixed-phase clouds, influencing their persistence and radiative properties. Knowledge of the abundance, sources, and nature of INPs over the remote Arctic Ocean is scarce, particularly in the Eurasian Arctic. In this work, we present summertime measurements of INP concentrations ($N_{\mathrm{INP}}$) in
immersion mode from the ship-based Arctic Century Expedition exploring the Barents, Kara, and Laptev Seas and the adjacent high Arctic islands and archipelagos during August to September 2021. Atmospheric $N_{\mathrm{INP}}$ were found to be lower than in continental high-latitude sites, particularly at temperatures below -15 °C, suggesting a lower abundance of mineral dust INPs. The geographical $N_{\mathrm{INP}}$ variability in the Eurasian Arctic shows that the highest $N_{\mathrm{INP}}$ are observed when the ship was in the ice-free ocean, marginal ice zones (MIZ), and in the vicinity of land. Very low $N_{\mathrm{INP}}$ were measured within the ice pack. The peak
$N_{\mathrm{INP}}$ was observed north of Novaya Zemlya where backward trajectories indicate air parcels arriving from the western Siberian coast. Overall, we find that INP sources are local to regional, with little evidence for long-range transport to the investigated area of the Eurasian Arctic in summer months.

## 1 Introduction

The Arctic warming mechanism is intricately linked to the presence of ice in mixed-phase clouds (MPCs). The phase par-
titioning of hydrometeors in Arctic low-level MPCs affects the Arctic's radiation budget (Korolev et al., 2017; Serreze and Barry, 2011) through cloud phase feedbacks, which is a decrease in cloud albedo upon glaciation (Tan and Storelvmo, 2019). Primary ice formation in MPCs occurs on ice-nucleating particles (INPs), capable of catalyzing ice nucleation at temperatures above -38 °C, below which cloud droplets freeze homogeneously (Bigg, 1953; Vali et al., 2015). Cloud glaciation alters the cloud optical thickness and lifetime, thereby affecting the surface energy balance by modulating the reflection of sunlight and
trapping of outgoing longwave radiation. Murray et al. (2021) highlighted that an accurate representation of INPs in climate models is essential for predicting the microphysical and radiative properties of Arctic MPCs in the future. To this end, the scarcity of observations and the uncertainties in abundance and sources of INP remain a challenge. As the Arctic continues to warm, changes in INP sources, caused by increased emissions from open water, increasing wind speed and wave height,





or increased biological activity, are expected to change INP abundance and thereby cloud properties, which enhances positive
feedback mechanisms and exacerbates regional warming (Murray et al., 2021).

Previous efforts to measure the abundance, variability, sources, and origins of INP in the Arctic have shown that both
terrestrial and marine aerosols can serve as INPs in this region (Hartmann et al., 2021). While terrestrial sources of mineral
dust INPs are less prominent compared to the mid-latitudes, they still contribute notably through high-latitude dust emitted
from, e.g., coastal Greenland (Li et al., 2023), Siberia (Porter et al., 2022), glacial outwash plains in Svalbard (Tobo et al.,
2019), and Iceland's deserts (Sanchez-Marroquin et al., 2020). Creamean et al. (2018) and Irish et al. (2019) found a positive
correlation between $N_{INP}$ measured on ships and the duration that sampled air masses spend over land, underscoring the
dominance of terrestrial sources. Additionally, terrestrial sources of biogenic INPs have been associated with sediments from
rivers (Tobo et al., 2019), vegetated regions (Conen et al., 2016), and thawing permafrost (Barry et al., 2023; Creamean et al.,
2020). In the marine environment, deposited dust on the water surface can be re-suspended to the atmosphere during sea spray
aerosol (SSA) generation (Cornwell et al., 2020). Additionally, marine biogenic aerosols (MBAs) from sea spray (DeMott et
al., 2016; Wilson et al., 2015; Bigg, 1996), including marine organics (McCluskey et al., 2018; Wilson et al., 2015), bacteria,
and fragments of marine organisms (Šantl Temkiv et al., 2019; Bigg and Leck, 2001), phytoplankton exudates (Ickes et al.,
2020; Hartmann et al., 2020; Creamean et al., 2019), and marine diatoms (Knopf et al., 2011), have all been suggested as
effective INPs, particularly at temperatures above -15 °C (Murray et al., 2012). Collectively, previous findings indicate that
the INP population in the remote Arctic is a mixture of aerosol from both the local terrestrial and marine environments, with
a possible contribution from long-range transport (Murray et al., 2021). Previous studies (Bigg, 1996; Creamean et al., 2018;
Schmale et al., 2021) also show that INP levels in the Arctic vary seasonally, with higher concentrations typically observed
during the summer months at the same time as increased biological activity and terrestrial dust emissions.

The Arctic is particularly susceptible to climate change and has experienced accelerated warming over the past few decades
(Forster et al., 2021). Notable evidence of climate change in the Arctic includes the perennial retreat of sea ice cover in all
seasons (Wendisch et al., 2019). Future warming associated with receding Arctic sea ice will cause a strong positive surface-
albedo feedback (Hall, 2004). With more open water, wind-induced SSA generation is expected to increase (Browse et al.,
2014). Additionally, Gabric et al. (2018) demonstrated a significant rise in MBA burden connected to the declining sea ice
extent, induced by a concomitant elevation in the primary production of phytoplankton due to increased light availability
(Galindo et al., 2016), warming of the ocean mixed layer (Gabric et al., 2005), and enhanced nutrient supply (Becagli et al.,
2011). It is imperative for regional climate models to incorporate the dynamic changes in sea ice, MBA emission, detailed
cloud microphysics, and cloud radiative feedback to simulate the future climate in the Arctic.

In this work, we present ship-based INP measurements from the Arctic Century Expedition over the previously unexplored
Barents, Kara, and Laptev Seas and the adjacent high Arctic islands and archipelagos in the Eurasian Arctic. We report on
the current state of INP abundance, spatiotemporal variability, source regions, and origins to improve the understanding of
atmospheric INPs over the remote Eurasian Arctic Ocean.



# 2 Data and methods

## 2.1 Campaign overview

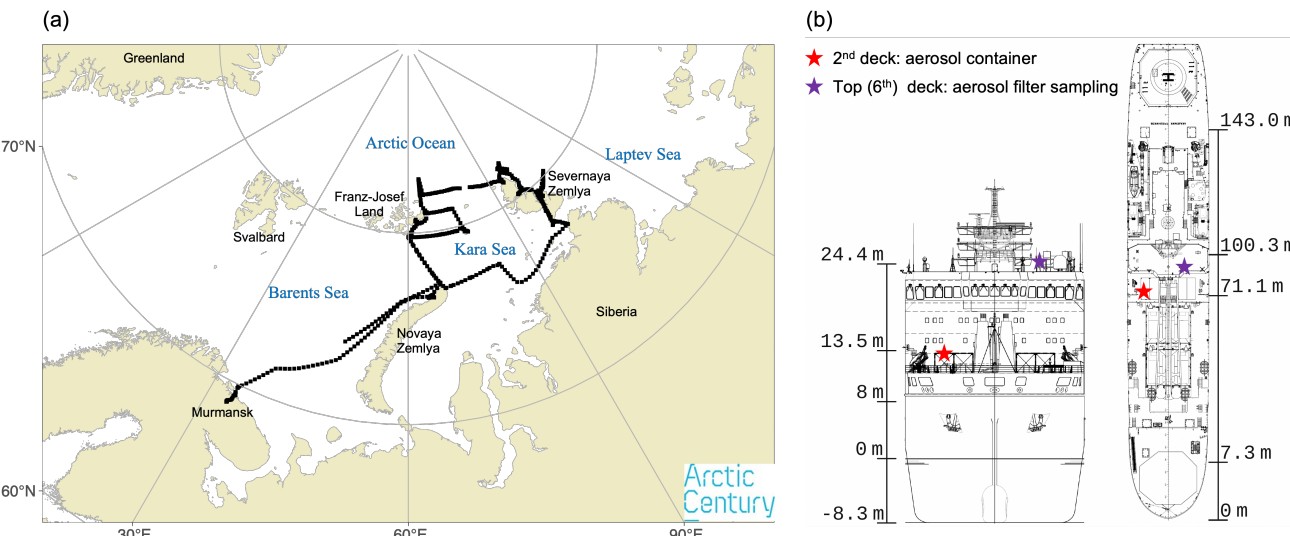

**Figure 1.** (a) Map of Arctic Century Expedition ship track. The cruise departure and return point was the harbor of Murmansk. The black squares show the hourly ship position during the campaign. Location information is missing at the beginning of the campaign due to restrictions from the local authority. (b) The equipment location on board the *RV Akademik Tryoshnikov* (adapted from vessel plans by the Arctic and Antarctic Research Institute). Height is provided relative to the approximate water line in the front view (left panel), and distance from the ship's bow in the top view (right panel). Further information on the instrumentation in both sectors is given in Table 1.

The Arctic Century Expedition took place from 5 August to 6 September 2021. Collocated measurements of atmospheric and marine physics and chemistry were conducted on the research vessel (RV) *Akademik Tryoshnikov*. Figure 1 (a) shows the route of the expedition. The Arctic Century Expedition started and ended at the harbor in Murmansk, Russia (68.98 °N, 33.09 °E) and explored an extensive area in the Eurasian Arctic Seas, including rarely accessible locations in the Kara and Laptev Seas, and the archipelagos of Franz-Josef Land, Novaya Zemlya, and Severnaya Zemlya.

A comprehensive set of atmospheric aerosol sampling and measurement (online and offline) was conducted on board. The locations of the measurement set-ups are indicated in Fig. 1 (b), and instrumentation at each location is given in Table 1. On the 2nd deck, monitoring and sampling of ambient aerosol were conducted from an aerosol container laboratory set-up following the configuration described in Li et al. (2025). A combination of online INP measurements and sampling for offline INP analysis was used to quantify $N_{\text{INP}}$. Additionally, the aerosol concentration and number size distribution were monitored continuously. On the top deck (6th deck), filter samples were collected for INP analysis after the expedition. Details on instrumentation, sample collection, and analysis are provided below.





**Table 1.** Summary of instrumentation set-up on the RV *Akademik Tryoshnikov* during the Arctic Century Expedition (for onboard location see Fig. 1). Measurement principles are provided in Section. 2.2. The abbreviations of instruments represent Horizontal Ice Nucleation Chamber (HINC, described in Lacher et al., 2017), Scanning Mobility Particle Sizer (SMPS), Aerodynamic Particle Sizer (APS), Condensation Particle Counter (CPC), and Low-Volume Sampler (LVS).

| Onboard location | Instrument | Temporal resolution | Function |
|---|---|---|---|
| Aerosol container laboratory (2nd deck) | HINC | 20 min | INP measurement (online) |
| | Impinger | 3 h | Aerosol collection in water for offline INP analysis |
| | SMPS | 4 min | Particle size distribution (0.012 - 0.6 µm) |
| | APS | 4 min | Particle size distribution (0.5 - 20 µm) |
| | CPC | 1 s | Particle total concentration |
| Top (6th) deck | LVS with $PM_{10}$ inlet | 12 h | Aerosol collection on filters for offline INP analysis |

## 2.2 Sample collection

### 2.2.1 Ambient aerosol samples

From the aerosol container laboratory, ambient aerosols were collected 3 times a day, for 3 hours each, into 15 mL ultra-pure water (W4502-1L, Sigma-Aldrich), using the high flow-rate impinger (Coriolis® μ, Bertin Instruments, with a lower limit aerodynamic cut-off size of 0.5 µm) at a flow rate of 300 L min$^{-1}$. Additional ultra-pure water (W4502-1L, Sigma-Aldrich) was constantly supplied to the sampling container during the operation of the impinger via a refilling system to compensate for evaporation loss. On the top deck, aerosol particles were collected onto 47 mm polycarbonate membrane filters (Whatman, 0.4 µm pore size) using a low volume sampler (LVS, Model DPA14, Digitel) with a 10-µm particulate matter ($PM_{10}$) inlet that excludes particles that are larger than 10µm in diameter from being collected (samples are hereafter referred to as $PM_{10}$ filters). The LVS inlet was approximately 25 m above sea level. The operating flow rate was maintained at 38.3 L min$^{-1}$ for 12-hour sampling intervals.

The impinger samples and $PM_{10}$ filters were stored at -20 °C on board, for transport, and after the campaign at the ETH laboratory until analysis. During the campaign, a total of 75 impinger and 50 $PM_{10}$ filter samples were collected.

## 2.3 INP analyses

### 2.3.1 Impinger and filter samples

Impinger samples were brought out of a freezer into the fridge at 4 °C overnight before analysis. Membrane filter samples were immersed in 15 mL ultra-pure water (W4502-1L, Sigma-Aldrich) and agitated using a sonicator for 30 min to re-suspend the particles from filters into the water. The impinger and filter suspensions were subsequently used for immersion-mode INP analysis with DRINCZ (David et al., 2019).




### 2.3.2 DRoplet Ice Nuclei Counter Zurich (DRINCZ)

Each liquid sample was pipetted into a Polymerase Chain Reaction (PCR) tray with 96 aliquots of 50 μL and cooled in an ethanol bath at 1 °C min⁻¹. Freezing events were detected optically from the change in transparency of an aliquot upon freezing. $N_{INP}$ were derived at each integer temperature following Vali (1971):

$$N_{INP}(T) = -\frac{\ln\left[1 - \dfrac{N_{frz}(T)}{N_{tot}}\right]}{V_{aliquot}} \cdot \frac{V_{water}}{V_{flow}} \tag{1}$$

where $N_{INP}(T)$ is the INP concentration at temperature $T$, $N_{frz}(T)$ is the number of frozen aliquots at temperature $T$, $N_{tot}$ is the total number of aliquots ($N_{tot} = 96$), $V_{aliquot}$ is the aliquot volume ($V_{aliquot} = 50$ μL), $V_{water}$ is the total water volume of the Coriolis sample or the volume of water used to suspend $PM_{10}$ filters. $V_{flow}$ is the sampled air volume. Field blank samples, undergoing the same procedures as actual samples, were collected every three days during the campaign. The $N_{INP}$ were corrected for the background of field blank samples by subtracting the differential INP spectrum of field blanks from samples (Vali, 2019). Based on the limit of detection (LOD) of DRINCZ and the purity of the nano-pure water, the highest temperature for $N_{INP}$ detection was approximately -5 °C (above which sampled air volumes are too small to detect lower concentrations), and the lowest temperature at which $N_{INP}$ can be reliably reported was -25 °C (below which nano-pure water starts to freeze). The overall uncertainty of the reported freezing temperatures is ±0.9 °C (David et al., 2019).

### 2.3.3 Horizontal ice nucleation chamber (HINC)

To extend observations of the INP spectrum to low temperatures, measurements were conducted with HINC (Lacher et al., 2017). HINC was operated alternately at $T$ = -30 °C and $T$ = -34 °C (± 0.4 °C), at a relative humidity with respect to water of $RH_w$ = 104 % (± 1.5 %), representative for immersion-mode ice nucleation conditions. The two experimental temperatures were alternated after half a day when the moisture source inside HINC was depleted, and the measurement had to be restarted. Details on the field configuration of HINC can be found in Li et al. (2022), and operational details of detecting and distinguishing ice crystals from droplets are given in Lacher et al. (2017). To account for background ice crystal counts from frost particles detaching from the inner chamber surface, a period of filtered air measurement (5 min) before and after each sampling interval (15 min) was included in the measurement sequence. The background count of ice crystals and the limit of detection are determined based on Poisson statistics. The LOD is given by the mean + $1\sigma$ of the background counts. $N_{INP}$ was calculated by subtracting the mean background counts from the ice counts during the sampling interval (see Lacher et al., 2017 for details). During the Arctic Century Expedition, 285 $N_{INP}$ out of 589 measurement intervals were above the LOD of the instrument. In other words, the 285 $N_{INP}$ data points have a significance level of 68.3 % ($1\sigma$), which we considered reliable based on the limitations of the instrument at the measurement conditions.





## 2.4 Supporting measurements and analyses

### 2.4.1 Particle size distribution

From the aerosol container laboratory on the second deck, the size distribution of submicron particles was measured using a scanning mobility particle sizer (SMPS, Model 3938, comprising a 3088 soft X-ray neutralizer, a 3082 classifier, a 3081 long differential mobility analyzer, and a 3787 CPC, TSI Inc.). The sampling flow rate of the SMPS was 0.6 L min$^{-1}$ with a sheath-to-sample ratio of 10:1, leading to an observable size range from approximately 12 to 600 nm in electrical mobility diameter. A multiple charge correction was applied to account for the misclassification of larger particles carrying multiple charges.

Parallel to the SMPS, the size distribution of coarse-mode particles (ranging from approximately 0.5 to 20 μm in aerodynamic diameter) was measured by an aerodynamic particle sizer (APS, model 3321, TSI Inc.) at a flow rate of 1 L min$^{-1}$. The SMPS and APS were operated at the same time resolution of 4 min to align the obtained size distributions. The electrical mobility diameters obtained from the SMPS and aerodynamic diameters from the APS were converted to volume-equivalent diameters assuming an average particle density of 2 g cm$^{-3}$ (Tunved et al., 2013). The surface area of particles was calculated assuming

a sphere shape at all sizes. An additional CPC (Model 3787, TSI Inc.) was used to monitor the total aerosol particle number concentration continuously.

### 2.4.2 Chemical composition analysis

Inductively coupled plasma-optical emission spectrometry (ICP-OES, Model 5100, Agilent Technologies) was used to detect 11 selected elements (Al, Ca, Cl, Fe, K, Mg, Mn, Na, P, S, Si) in the 75 aerosol suspension samples collected by the impinger.

The impinger samples were diluted by a factor of 10 with 2 % HNO$_3$ solution prior to the chemical analysis. Quality control was established by the measurement of blank samples and standard reference materials of each element processed in parallel (see details of experimental protocols in Gilli et al., 2018). The resulting elemental compositions for Cl, Fe, Mn, and K were below the LOD and thus are not discussed. The elements analyzed in this work include P, S, and joint classes of AlSiCa and NaMg, representing dust and sea salt components, respectively, according to the standards introduced in Hiranuma et al.

140 (2013).

### 2.4.3 Meteorological and sea ice conditions

An automated weather station (model AWS420, Vaisala) was operated on the top deck, delivering measurements of ambient pressure at 20 m a.s.l., air temperature and relative humidity at 23.7 m a.s.l., and relative and absolute wind speed and direction at 30 m a.s.l. The recorded measurements were processed automatically by the Vaisala software. In addition, relative wind

speed and directions were measured from the wind sensor (Model WXT532, Vaisala) mounted on the LVS at approximately 26 m a.s.l.

For comparison to in situ observations at the position of the RV *Akademik Tryoshnikov*, the sea ice concentrations were determined from the daily observations of the U.S. National Ice Center. Additionally, sea ice coverage from hourly ERA5 data




(5th generation of ECMWF atmospheric reanalysis of the global climate covering the period from January 1950 to the present,
Hersbach et al., 2020) with a 30 km horizontal resolution was used to characterize the sea ice conditions. The sea ice coverage
was classified following Aksenov et al. (2017) and Strong and Rigor (2013), where regions with < 15 % are defined as the
(ice-free) ocean, sea ice coverage between 15 % and 80 % as the marginal ice zone (MIZ, the transitional zone between open
sea and dense ice pack), and > 80 % sea ice as the ice-pack.

### 2.4.4 Backwards trajectory analysis

Air parcel backward trajectories were computed to identify the origin of the air masses reaching the ship's location using
the Lagrangian analysis tool LAGRANTO (Sprenger and Wernli, 2015; Wernli and Davies, 1997). Two-day (48-h) backward
trajectories were calculated using the three-dimensional wind fields from the hourly ERA5 data with a horizontal resolution
of 0.5°. The trajectories were launched every hour, starting from the ship's position along the ship track, with pressure closest
to the sea level (1000 hPa). Trajectories were removed when they were above the modeled height of the boundary layer or
experienced precipitation (surface precipitation below air parcel position > 0.1 mm/h), since this study focuses on the aerosol
sources from the boundary layer. The trajectories were categorized according to the over-passed surface types (sea ice coverage
or land).

## 3  Results and discussion

### 3.1  Temperature-dependent variability of Arctic INP concentrations

The cumulative $N_{INP}$ as a function of freezing temperature from both online and offline measurements using HINC ($T$ = -34
°C and -30 °C), impinger samples, and PM$_{10}$ filters ($T \geq$ -25 °C) is shown in Fig. 2. Across the assessed freezing temperatures,
$N_{INP}$ spans 1 to 3 orders of magnitude, overlapping the parameterization derived from measurements in Svalbard (Li et al.,
2022) except impinger samples. The discussion on the difference in $N_{INP}$ measured across HINC, impinger samples, and PM$_{10}$
filters was detailed in Appendix A. Figure 2 shows a comparison of our observed $N_{INP}$ from the Arctic Century Expedition
to the range of $N_{INP}$ derived from midlatitude precipitation samples (Petters and Wright, 2015, the range between the light
magenta lines). Overall, $N_{INP}$ observed during the Arctic Century Expedition is up to two orders of magnitude lower compared
to the $N_{INP}$ at mid-latitudes, especially at T < -15 °C. The significantly lower $N_{INP}$ at $T$ < -15 °C, where mineral dust becomes
a dominant source of INPs (Hoose and Möhler, 2012), indicates that the concentration of mineral dust INPs is lower over the
Eurasian-Arctic Ocean compared to mid-latitudes. Smaller differences in $N_{INP}$ were found at $T$ > -15 °C, in particular for $T$
> -10 °C, where $N_{INP}$ from the Arctic Century Expedition aligned with the range reported by Petters and Wright (2015). For
a comparison within different Arctic regions, Fig. 2 also includes ship-based INP observations in the summer over the Arctic
Ocean from previous work (Hartmann et al., 2021; Bigg, 1996; DeMott et al., 2016; Bigg and Leck, 2001; Welti et al., 2020;
Creamean et al., 2019). Our offline $N_{INP}$ are similar to $N_{INP}$ reported by Hartmann et al. (2021). Conversely, the $N_{INP}$ measured
with HINC at $T$ = -34 °C and -30 °C were systematically lower than the low-temperature observations during this summer




measurement campaign circumnavigating Svalbard. A difference in the concentration of mineral dust INP could be the reason.
Hartmann et al. (2021) reported an abundance of mineral dust from Greenland and adjacent Svalbard, following ice and snow
melt (see (Tobo et al., 2019) for information on Arctic dust sources). Other Arctic $N_{\text{INP}}$ measurements above -30 °C by Bigg
(1996); Bigg and Leck (2001); DeMott et al. (2016); Creamean et al. (2019); Welti et al. (2020) corroborate our observations.

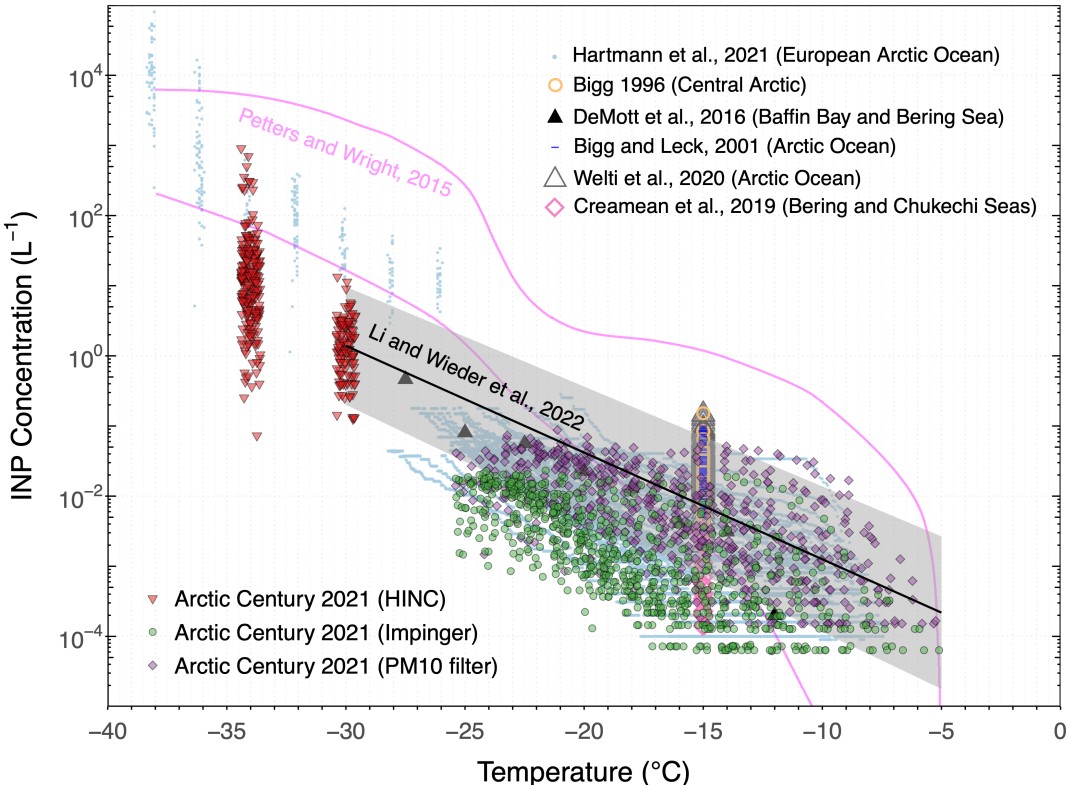

**Figure 2.** Temperature spectrum of $N_{\text{INP}}$ measured with HINC, Coriolis impinger and PM$_{10}$ filters (shown as filled triangles, circles and
diamonds, respectively). Previously observed $N_{\text{INP}}$ in the high Arctic Ocean reported in Bigg (1996); Bigg and Leck (2001); DeMott et al.
(2016); Creamean et al. (2019); Welti et al. (2020); Hartmann et al. (2021) are shown for comparison. The area between the two lines in light
magenta spans the range of $N_{\text{INP}}$ observed from precipitation samples collected in mid-latitudes. (Petters and Wright, 2015). The black solid
line and gray shaded area represent the INP parameterization derived from measurements in Svalbard (Li et al., 2022).

Overall, for $T$ < -15 °C, we observe lower $N_{\text{INP}}$ in the Eurasian Arctic compared to the mid-latitude range. We propose that
this is due to lower concentrations of dust. For $T$ > -15 °C, $N_{\text{INP}}$ is similar to the mid-latitude range because of the presence
of ice-nucleating MBAs over the Eurasian Arctic Ocean that match concentrations of biological INPs over continental mid-
latitudes.



## 3.2 Correlations of INP to aerosol number, size, and surface area

Figure 3 shows the correlations between $N_{\mathrm{INP}}$ and parameters related to particle size (see Table B1 for the correlation param-
eters). Among the investigated freezing temperatures, a significant, moderate correlation was exclusively observed at $T = -34$
°C for all parameters. At this temperature, the notable positive correlations between $N_{\mathrm{INP}}$, particle concentration, and surface
area across all sizes indicate that as the freezing temperature decreases and approaches the homogeneous freezing temperature,
a considerable fraction (1 in $10^2$) of all particles become IN active. Conversely, at higher temperatures, $N_{\mathrm{INP}}$ demonstrates a
mostly insignificant correlation with bulk aerosol properties (except the small negative correlation with $S_{>0.5}$). This decoupling
can be attributed to INPs constituting a small fraction of ambient aerosols - only about 1 in $10^5$ particles are IN-active at -15 °C.
The significant but weak anti-correlation between $N_{\mathrm{INP}}$ and $S_{>0.5}$ is unexpected, pointing towards higher $N_{\mathrm{INP}}$ when the particle
number concentration is low, which is the case when INPs come from a local source that has not been diluted by transport.
This suggests the contribution is local at warmer temperatures where bioaerosols are IN-active. However, we note that the
small dataset size limits a more comprehensive interpretation. Nevertheless, our findings in the maritime Arctic environment
challenge the established parameterization of ambient $N_{\mathrm{INP}}$ based on specific aerosol sizes (e.g., DeMott et al., 2010; Niemand
et al., 2012; DeMott et al., 2015; McCluskey et al., 2018).







**Figure 3.** Correlation of $N_{INP}$ measured from $PM_{10}$ filters and HINC with (a) the particle number concentration with volume-equivalent diameter larger than 0.5 μm ($n_{>0.5}$), (b) the particle number concentration with volume-equivalent diameter smaller than 0.5 μm ($n_{<0.5}$), (c) the surface area concentration of particles larger than 0.5 μm ($S_{>0.5}$) and (d) the surface area concentration of particles smaller than 0.5 μm ($S_{<0.5}$). All size distribution parameters are taken from the SMPS/APS measurements described in Section 2.4.1. Note that $n_{>0.5}$ and $S_{>0.5}$, which correlate with $N_{INP}$ measured at $T$ = -30 and -34 °C, only consider particle sizes of up to 2.5 μm, i.e., the upper size threshold of HINC. Colors represent $N_{INP}$ measured at 5 different temperatures indicated in the figure; dashed lines show the linear regression. The $r$ values in the figure indicate the correlation coefficients calculated with statistical significance ($p < 0.05$). The correlation coefficients and statistical significance for all correlations are given in Tab. B1 in the Appendix.





### 3.3 Chemical composition and sources of INPs over the Eurasian-Arctic Ocean

To further investigate the nature and source of INPs, the elemental composition of the ambient aerosol was determined from the impinger samples. Figure 4 exhibits the correlation analysis between $N_{INP}$ and multiple representative elements in the bulk aerosol samples. Statistically significant correlations were not found for $N_{INP}$ at -10 °C and -15 °C, likely due to the small ice active fraction of particles. In general, $N_{INP}$ is poorly correlated with the concentration of sea salt, which is expected since the soluble Na and Mg containing sea salt particles typically do not act as immersion INP. Despite the weak correlations at -10 and -15 °C, $N_{INP}$ increases with the concentration of AlSiCa (indicator for mineral dust), indicating an overall weak contribution of dust or covariant INP species over the Arctic Ocean. Additionally, a significant but weak correlation was found between $N_{INP}$ at -20 °C and phosphorus concentrations, and moderate but not significant correlations are observed between $N_{INP}$ and sulfur concentrations. In the biogeochemical cycle in the Arctic Ocean with minimal anthropogenic influence, phosphorus plays a crucial role as a nutrient for marine biology. Similarly, marine-released sulfur, often derived from dimethyl sulfide (DMS), a byproduct of marine phytoplankton and microbial metabolism (e.g., Becagli et al., 2019; Gabric et al., 2018), suggests that the appearance of marine biological activity, such as phytoplankton blooms, could enrich the INP population.





 

**Figure 4.** Correlation of $N_{\mathrm{INP}}$ with the concentration of indicator compounds measured from impinger samples using ICP-OES. Correlation of $N_{\mathrm{INP}}$ with (a) NaMg (sea salt indicator), (b) AlSiCa (dust indicator), (c) phosphorus (nutrient indicator) and (d) sulfur (DMS indicator) (not plotted at $T$ = -10 °C because only 2 data points were available). Colors represent $N_{\mathrm{INP}}$ measured at different temperatures indicated in the figure. Dashed lines show linear regression. The $r$ values displayed in the figure are the correlation coefficient calculated with statistical significance ($p < 0.05$). The correlation coefficients and statistical significance of all data are given in Tab. B2.

**3.4   Geographical variability of INP concentrations across the Arctic marine and ice landscape**

Figure 5 shows the geographical variability of $N_{\mathrm{INP}}$ measured along the ship track with HINC and PM$_{10}$ filters. Additional maps for $N_{\mathrm{INP}}$ measured from the impinger samples are shown in Fig. D1 in Appendix D. Generally, there is a tendency that at higher latitudes, lower $N_{\mathrm{INP}}$ are observed at all temperatures, except PM$_{10}$ filters collected near the Severnaya Zemlya, highlighting the influence of nearby terrestrial sources. $N_{\mathrm{INP}}$ vary with sea ice coverage, with the highest $N_{\mathrm{INP}}$ often occurring over the





ice-free ocean or marginal ice zone (MIZ), particularly near land, and less so within the dense ice pack. Fig. 6 categorizes INP
         temperature spectra by sea ice cover. Observations at $T > -12\ °C$ over the ice pack showed $N_{INP}$ below the LOD. T-test results
         indicate that at $T \geq -20\ °C$, the mean $N_{INP}$ is systematically, but not significantly higher over the ice-free ocean than over the ice
         pack and higher in the MIZ than on the ice-free ocean. Mean $N_{INP}$ at -30 °C and -34 °C are significantly different over the ice-
         free ocean compared to within the ice pack or the MIZ. Potential sources for elevated $N_{INP}$ over ice-free waters, particularly in

proximity to coastal regions, include biogenic INPs originating from marine biota, such as phytoplankton exudates (Creamean
         et al., 2019), augmented by nutrient influx from the warmer, saltier Atlantic waters entering the Arctic Ocean during ice-melting
         seasons (a process known as Arctic Atlantification, Tuerena et al., 2022), and mineral dust input from river runoff and thawing
         permafrost (Tobo et al., 2019; Hartmann et al., 2021). An additional potential source of INP in the MIZ that has previously
         been reported is sea ice algae growing at the marginal or seasonal ice zone and in open leads (Gabric et al., 2018; Kirpes et al.,

2019).



**Figure 5.** Geographical variability during the Arctic Century Expedition of INP concentrations at (a) $T = -15$ °C and (b) $T = -20$ °C sampled with $PM_{10}$ filters and analyzed with DRINCZ; (c) $T = -30$ °C and (d) $T = -34$ °C measured with HINC (note different color scales). Different shapes represent categories of sea ice extent (indicated in the figure legend) when the measurements were taken based on the daily observation of sea ice concentrations. Note the difference in color scale when comparing at different temperatures.





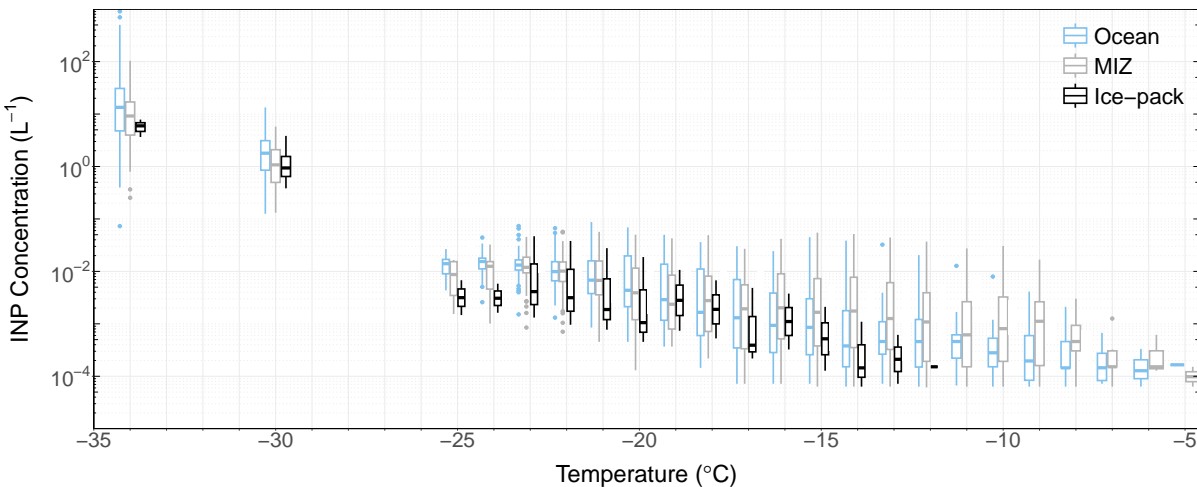

**Figure 6.** Cumulative INP temperature-spectra derived from HINC (-30 and -34 °C), impinger samples and $PM_{10}$ filters (-5 to -25 °C), given different sea ice conditions (sky blue: ice-free ocean; light grey: marginal ice zone (MIZ); black: ice pack). The lower and upper bounds of the boxes indicate the 25 % and 75 % quantiles, and the horizontal line within the boxes represents the median $N_{INP}$. Whiskers have a length of 1.5 × IQR (interquartile range), and individual dots mark outliers.





### 3.5 Influence of Air mass origins on INP variability

To investigate the variability in $N_{INP}$ with airmass origin, 2-day backward trajectories starting from the boundary layer at the ship's location during observation periods were calculated. Figures 7 (a) and (b) show the trajectories colored according to the observed $N_{INP}$ at $T$ = -34 °C and $T$ = -15 °C, respectively, at sea level. For both temperatures, the highest $N_{INP}$ were captured when the ship approached the northern coast of Novaya Zemlya, where the air masses originate from the western Siberian coast near the estuary of the Pyasina River. Local terrestrial sources from Novaya Zemlya or long-range aerosol transport may be significant contributors to these high-$N_{INP}$ cases. Porter et al. (2022) reported a similar air mass origin for the highest $N_{INP}$ active at temperatures above -15 °C. They measured in the central Arctic in 2018 during the summer season, suggesting the Novaya Zemlya region can be a strong Arctic INP source. Previous studies have reported that the shallow seas off the Siberian coast and Arctic archipelagos are heavily affected by fluvial discharge rich in organic matter, silt, clay, and nutrients (Ahmed et al., 2020; Juhls et al., 2019). Thawing permafrost in the summer enhances the mobilization of soil, nutrients, and active microbes into rivers and runoff into the ocean (Hultman et al., 2015). Aerosolization of silt, clay, and soil components at the water-air interface may contribute to the $N_{INP}$ at $T$ = -34 °C. Elevated marine biological productivity due to increased nutrient availability from fluvial input could have enhanced $N_{INP}$ at $T$ = -15 °C. The INPs are likely locally generated dust and biological particles is further supported by the chemical composition time series shown in Fig. C1 (Appendix C), where at the corresponding ship location around 03/09, high concentrations in all categorized compositions were observed. Further evidence that local sources are mainly contributing to the $N_{INP}$ at -34 °C, comes from moderately elevated $N_{INP}$ when the ship approached Franz-Josef Land and Severnaya Zemlya (Figures 7a). In these locations, air masses often originated from the central Arctic with high sea ice coverage, where the INP population is scarce, indicating local influences may dominate the INP source over long-range transport. Similarly, at $T$ = -15 °C, a few high INP occurrences coincided with air masses originating from the MIZ or ice packs, where the ship was located over the open ocean close to Severnaya Zemlya (Figures 7b). Local aerosol emissions from the ocean and the island likely contributed to the high INP population at this temperature.

Figures 7 (c) and (d) display 2-day backward trajectories classified by the overpassed types of surface during air mass history. In addition to the western Siberian coast, air masses containing moderate to high $N_{INP}$ also passed over ice-free ocean and land. In contrast, air mass trajectories passing over the ice pack or MIZ contained lower $N_{INP}$. One factor could be the absence of aerosolization via wave breaking and bubble bursting where there is sea ice. A correlation analysis between $N_{INP}$ and the residence time of air masses over different types of surface is shown in Fig. 8, with correlation coefficients provided in Tab. B3 in the Appendix. Negligible to no correlations were found between $N_{INP}$ and the percentage of time that trajectories spent in the boundary layer over the four different types of surface (i.e., ice-free ocean, land, MIZ, or ice pack), suggesting no clear pattern emerges, which surface types are most important for high $N_{INP}$. The missing correlation between surface types along trajectories and $N_{INP}$ along the ship track indicates that local INP sources are important in shaping INP population in the Eurasian-Arctic Ocean. From the combination of all these observations, it can be concluded that long-range transport generally plays a less important role in the concentration of INPs over the Eurasian-Arctic Ocean during the summer season than local emissions.



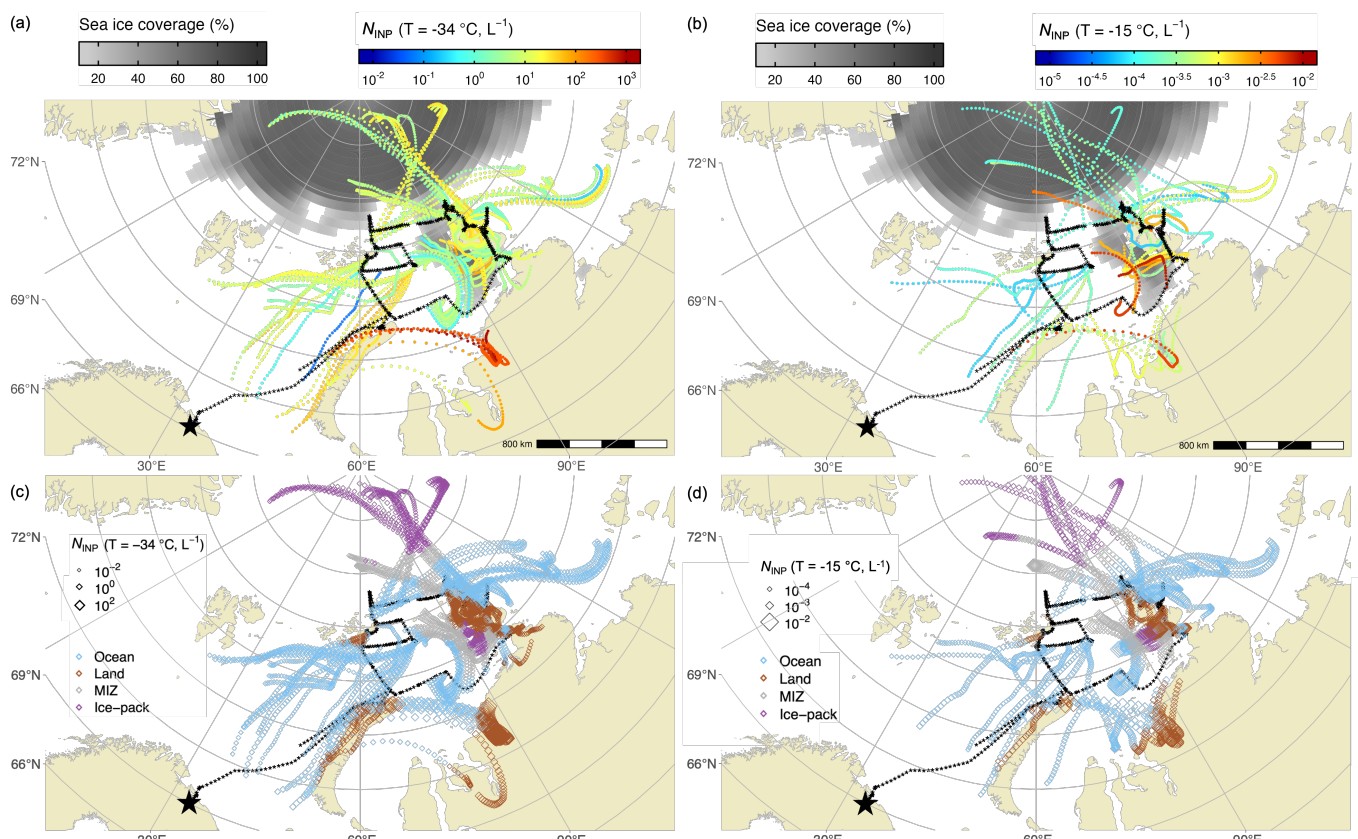

**Figure 7.** Two-day backwards trajectories starting from the ship location at sea level. A trajectory was launched every hour, during each INP sampling period. Points along the trajectories are marked at hourly intervals. The black asterisks represent the ship's track and the starting point of the back trajectories. The large black star marks the starting and ending point of the expedition (i.e., the port of Murmansk). The trajectories are colored by the $N_{INP}$ concentrations at (a) $T$ = -34 °C and (b) $T$ = -15 °C (note the different color scales for different temperatures). The sea ice coverage is shown in grayscale. The same backward trajectories are shown for measurements at (c) $T$ = -34 °C and (d) $T$ = -15 °C, but here colored by the four surface types passed over by the air parcel. The marker size indicates the $N_{INP}$.

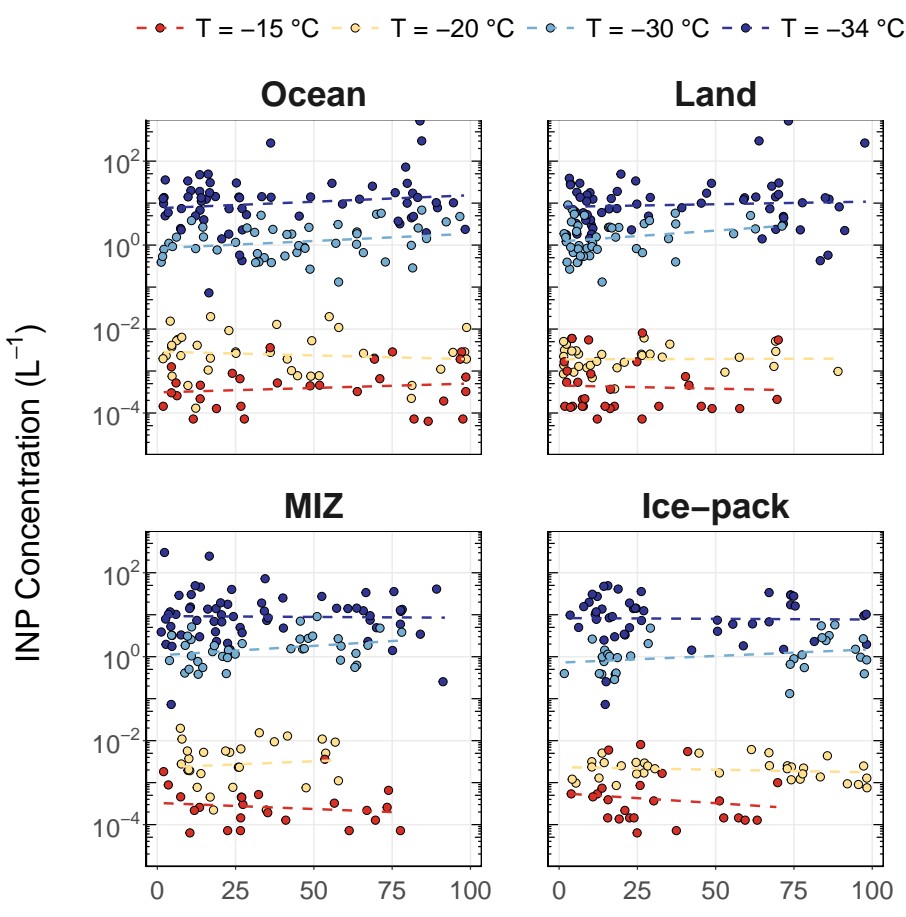

**Figure 8.** $N_{INP}$ as a function of the percentage of time that the 2-day backwards trajectories spent in the boundary layer over different types of surface. Colored circles represent $N_{INP}$ measured at four different temperatures; dashed lines show the linear regression of the corresponding data. None of the datasets shows an effect on $N_{INP}$ from the time air masses spent over land, ocean, or ice shelf. The correlation coefficients and statistical significance metrics are given in Table B3 in the Appendix.



### 265 3.6 Case study

In this Section, two selected periods of low and high $N_{\mathrm{INP}}$ during the campaign are discussed in detail. The analysis includes measurements of aerosol size, elemental analysis, and air mass history. The periods are selected based on measured $N_{\mathrm{INP}}$ at $T$ = -20 °C, where $N_{\mathrm{INP}}$ were in the bottom and top 25 % quartiles for the low and high $N_{\mathrm{INP}}$ period, respectively, and where parallel aerosol characterization was available (see Fig. A1 in the Appendix for $N_{\mathrm{INP}}$ time series). The ship's location during

the two periods and back trajectories of the sampled air are shown in Fig. 9. The low $N_{\mathrm{INP}}$ period occured when the ship sailed northeast of Franz-Josef Land at the edge of the MIZ, with the majority of air parcels originating from the northeastern Arctic ice pack, where the sea surface is mostly covered by ice and lacked terrestrial and marine aerosol sources. In contrast, during the high $N_{\mathrm{INP}}$ period, the ship was sailing along the western coast of Novaya Zemlya in the ice-free ocean, with air parcels rich in SSA, dust, and biogenic particles arriving from the west Siberian coast.

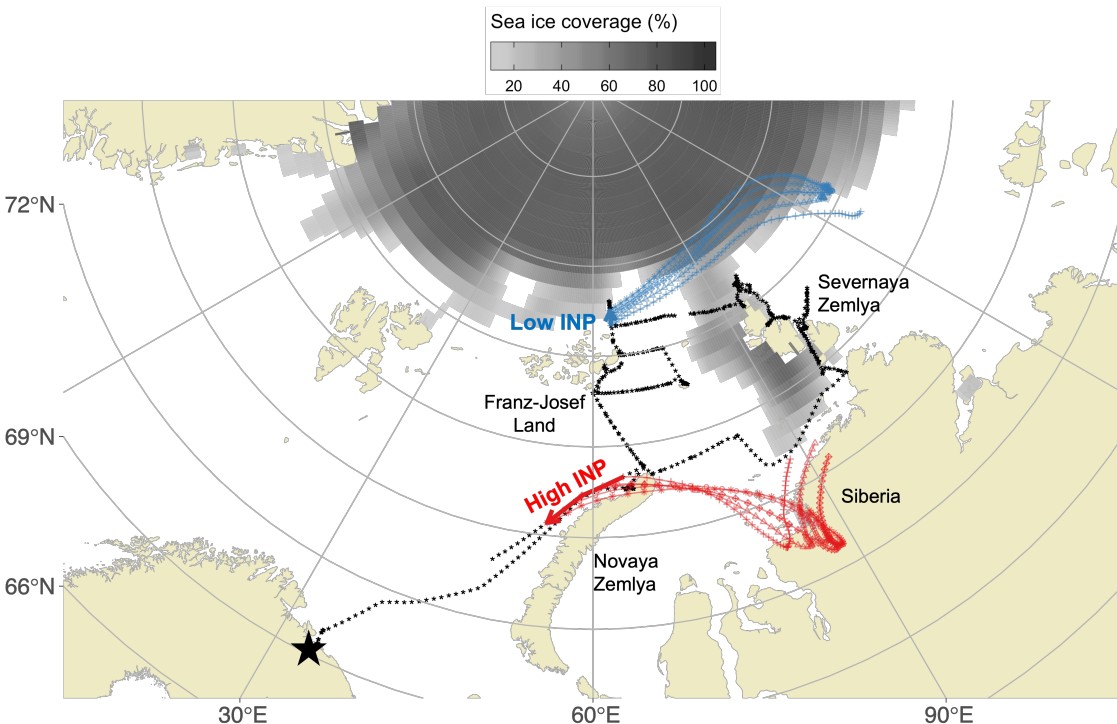

**Figure 9.** 2-day backwards trajectories for periods of low (blue) and high (red) $N_{\mathrm{INP}}$ cases ($T$ = -20 °C) during 12:00 - 20:00 UTC on 16/08/2021 and 07:00 - 20:00 UTC on 03/09/2021, respectively. The trajectories start from the ship's location at sea level and are launched hourly, with the points along the trajectory indicating hourly intervals. The solid thick arrows indicate the ship's track during the selected period. The black asterisks represent the ship's track, and the large black star marks the port of Murmansk, the start and end point of the expedition.



Figure 10 (a) shows the average particle size distribution during the time windows of the two periods. Notably, particle concentrations were three orders of magnitude higher across all size ranges during high $N_{INP}$. Particularly, aerosol particles with diameters larger than 5 μm were only detected during the high INP period and could be responsible for the elevated $N_{INP}$ measured from impinger and PM$_{10}$ filters. For particles below 2.5 μm captured by HINC, the high concentration in this size range could have caused the high $N_{INP}$ at -30 °C and -34 °C. Figure 10 (b) shows the abundance of tracer elements detected

in impinger samples to support the case study. Sea salt concentrations were over 100 times higher in the high $N_{INP}$ sample compared to the low $N_{INP}$ sample, likely due to less saline seawater and ice covering the sea-air interface in the low $N_{INP}$ case. Additionally, the average wind speed at the sea surface was much higher (18 to 24 m s$^{-1}$) in the high $N_{INP}$ period than during low $N_{INP}$ (3 to 6 m s$^{-1}$) due to the passage of a cyclone during the high INP case. Since the concentration of local sea spray aerosols strongly depends on wind-induced wave breaking and bubble bursting (Moallemi et al., 2024; Lewis et al.,

2004), a heavier aerosol load, carrying IN-active MBA can be expected during high $N_{INP}$. The higher phosphorous and sulfur contents during high $N_{INP}$ support the hypothesis of fluvial-marine transport of nutrients from meltwater runoff adjacent to Novaya Zemlya. Furthermore, approximately 25 times more AlSiCa (indicator of mineral dust) was detected in the high $N_{INP}$ sample compared to the low $N_{INP}$ sample. Albeit long-range transport of mineral dust cannot be ruled out, the local dust sources likely dominate as indicated by the concurrent elevated mineral dust and sea salt concentrations (see Figs. 10b, and C1

in the Appendix). The local dust sources can include re-suspended dust previously deposited at the ocean surface (Cornwell et al., 2020), aerosolization of muddy water surrounding the island, or wind-blown dust from the Novaya Zemlya coast due to high wind speed. In summary, differences in aerosol concentrations and the different air parcel origin led to a higher INP concentration during high winds from a southeasterly direction. The variation in strength of local island or ocean sources, as well as long-range transport of INP from the Siberian coast, could explain the difference in $N_{INP}$ for the two periods.



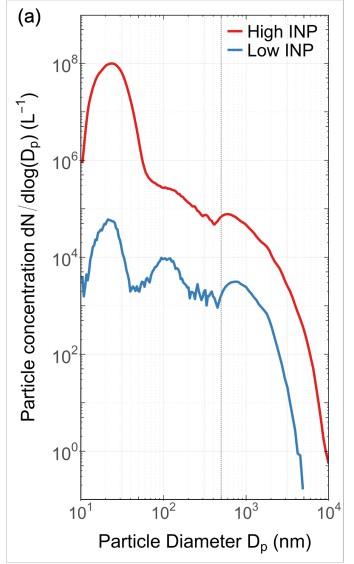
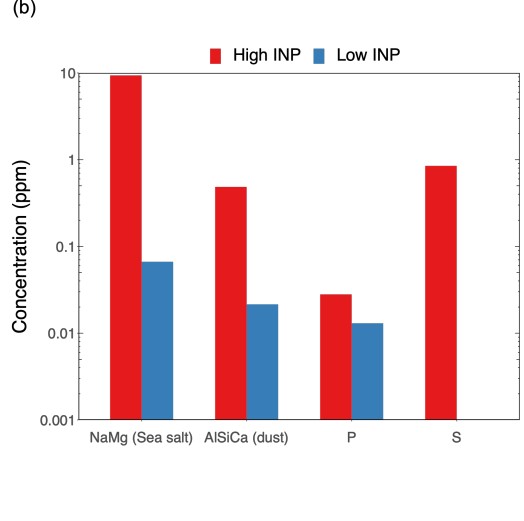

**Figure 10.** Physicochemical properties of aerosols for low and high $N_{INP}$ cases during 12:00 - 20:00 UTC on 16/08/2021 and 07:00 - 20:00 UTC on 03/09/2021, respectively. (a) Averaged particle size distribution. The vertical dashed line indicates the volume equivalent diameter of 500 nm, where SMPS and APS measurements overlap; (b) Concentrations of tracer elements measured by ICP-OES from impinger samples.

## 4 Summary and Conclusions

This study reports summertime observations of $N_{INP}$ over the Barents, Kara, and Laptev Seas in the Eurasian Arctic from August to September 2021. A combination of online and offline INP measurement techniques was deployed to cover a broad temperature range of immersion freezing temperatures from 0 to -34 °C and to investigate the spatiotemporal variability of INPs of different size ranges.

Atmospheric $N_{INP}$ in the summer Eurasian-Arctic was observed to be up to two orders of magnitude lower than in mid-latitudes, with concentrations varying by up to three orders of magnitude at individual freezing temperatures between -5 to -30 °C, and up to four orders of magnitude at -34 °C. In the open ocean, MIZ and ice packs consistently lower $N_{INP}$ were observed compared to measurements close to coastal Arctic sites, particularly for temperatures below -15 °C, indicating that the terrestrial aerosols significantly influence $N_{INP}$ in the Arctic region. However, if $N_{INP}$ are compared to year-around datasets from coastal Arctic locations in Wex et al. (2019), a similar range of concentrations was observed, pointing to a similar interseasonal, distance to land dependent spatial, and wind direction related quotidian variability in $N_{INP}$. The influence of terrestrial sources for the current observations is further supported by observations of occasional spikes in $N_{INP}$ when the ship was close to Arctic islands or archipelagos, while lower $N_{INP}$ were recorded when the ship was within the packed ice. The observed dependence of $N_{INP}$ on sea ice cover and proximity to snow-free land is consistent with previous studies in the Arctic Ocean around Svalbard (Hartmann et al., 2021). For additional confirmation, environmental analyses of high- and low-$N_{INP}$ samples reveal that less sea ice cover, higher wind speed, and proximity to terrestrial INP sources are key factors related to





higher $N_{\mathrm{INP}}$. Nevertheless, long-range transport of air parcels from the west Siberian coast to the Barents Sea could also have contributed to an elevation in $N_{\mathrm{INP}}$.

The accelerated warming in the future Arctic will reduce sea ice cover, increase wind speeds, and enhance permafrost thawing, which may augment INP loading and consequently cause a shift in Arctic MPC glaciation temperatures. Further research on the interactions of such dynamic changes in future climates on INP abundance and cloud feedback would be beneficial to advance the assessment of Arctic sea-aerosol-cloud-climate interactions.

**Appendix A: Temporal variability of $N_{\mathrm{INP}}$ from different devices used during the Arctic Century Expedition**







**Figure A1.** $N_{INP}$ time series from 3-hour impinger samples, 12-hour PM$_{10}$ filter samples and HINC measurements from 06/08/2021 to 04/09/2021 at temperatures of: (a) -15 °C; (b) -20 °C; (c) -30 °C and d) -34 °C (note different y-axis scale). Measurements below the LOD are not shown in (a) and (b) but are displayed in hollow triangles in (c) and (d). Rectangular boxes mark the time windows for selected low and high $N_{INP}$ cases.

At most freezing temperatures, $N_{INP}$ from PM$_{10}$ filters was observed to be higher than that from impinger samples. This
difference could be influenced by the larger volume of air processed by the impinger (approximately 54 m$^3$) compared to the PM$_{10}$ filters (approximately 27.6 m$^3$), which shifts the detectable range to lower $N_{INP}$ for the impinger samples. The minimum detectable $N_{INP}$ with DRINCZ decreases as the air volume increases. Therefore, the $N_{INP}$ detectable from impinger samples is lower, which is particularly relevant at the highest freezing temperatures. Several other factors, listed below in no particular order, may also account for the differences in $N_{INP}$ measured from impinger and PM$_{10}$ samples.



– Aerosols captured by the impinger are directly immersed in water, potentially reducing the IN activity of certain particles, such as mineral dust (Perkins et al., 2020). Filter samples, due to their shorter water exposure before INP analysis, are less likely to undergo such degradation.

    – The freeze-thaw cycle of freezing impinger samples for storage at -20 °C before melting the samples for analysis could deactivate INPs (Beall et al., 2020).

– Size selectivity of the samplers could explain the variation in INP abundance. The impinger collects particles larger than 0.5 μm, while $PM_{10}$ filters also sample particles smaller than 0.5 μm. Additional sub-0.5 μm particles, such as biogenic macromolecules, may have played a role for higher $N_{INP}$ from filter compared to impinger samples, particularly at temperatures above -15 °C. Despite the findings that IN activity of mineral dust often scales with particle size (e.g., DeMott et al., 2015; Welti et al., 2009), some research has also revealed that biological, IN-active macromolecules of
smaller size, e.g., marine organics smaller than 200 nm, can be effective INPs (McCluskey et al., 2018; Wilson et al., 2015) which would not be present in the impinger samples.

    – The difference in instrument positions (i.e., impinger samples and $PM_{10}$ filters were collected on the second and sixth deck, respectively, see Fig. 1b) could have contributed to a sampling bias. Previous observations of wind speed on the RV Akademik Tryoshnikov indicated that the ship's superstructure distorts the airflow, potentially introducing a bias in
measurements at various onboard locations (Landwehr et al., 2020). Such bias might similarly influence aerosol and INP measurements, with their abundance affected by the strength and direction of the ambient airflow.

Compared to the offline measurements, $N_{INP}$ measured with HINC at -34 °C and -30 °C generally align better with the values from the $PM_{10}$ filters than those from impinger samples. The observed alignment of $N_{INP}$ from HINC with those from $PM_{10}$ filters could come from HINC also sampling particles with sizes below 2.5 μm (Li et al., 2022; Lacher et al., 2017), therefore
covering the sub-0.5 μm particles like the $PM_{10}$ filters.

**Appendix B: Correlations of bulk aerosol parameters and air parcel history to $N_{INP}$**





**Table B1.** Pearson correlation coefficients (r) calculated between $N_{INP}$ (from $PM_{10}$ filters) at selected freezing temperatures and concentrations of different particle size-resolved parameters derived from SMPS and APS measurements. n is the number concentration and S the total surface area, separated for particles larger than 0.5 µm or smaller than 0.5 µm. $r$ values in bold text represents results with statistical significance ($p < 0.05$). $r$ values with $*$ denote moderate correlations ($0.3 < |r| < 0.7$), and with $**$ indicate strong correlations ($|r| > 0.7$). Statistical significant $r$ values ($p < 0.05$) are also displayed in Fig. 3.

| Parameter | $N_{INP}$ (T = -10 °C) | $N_{INP}$ (T = -15 °C) | $N_{INP}$ (T = -20 °C) | $N_{INP}$ (T = -30 °C) | $N_{INP}$ (T = -34 °C) |
|---|---|---|---|---|---|
| $n_{>0.5}$ | -0.269 | -0.089 | 0.021 | 0.064 | **0.579**$^*$ |
| $n_{<0.5}$ | -0.339$^*$ | -0.103 | 0.182 | 0.038 | **0.524**$^*$ |
| $S_{>0.5}$ | **-0.398**$^*$ | -0.202 | -0.108 | 0.031 | **0.506**$^*$ |
| $S_{<0.5}$ | -0.296 | -0.069 | 0.185 | 0.073 | **0.535**$^*$ |

**Table B2.** Pearson correlation coefficients (r) calculated between $N_{INP}$ at selected freezing temperatures and concentrations of elemental compositions indicating sea salt, dust, and marine biological sources were measured using ICP-OES. $r$ values in bold represent results with statistical significance ($p < 0.05$). $r$ values with $*$ denote moderate correlations ($0.3 < |r| < 0.7$), and with $**$ indicate strong correlations ($|r| > 0.7$). Statistical significant $r$ values ($p < 0.05$) are also displayed in Fig. 4.

| Composition | $N_{INP}$ (T = -10 °C) | $N_{INP}$ (T = -15 °C) | $N_{INP}$ (T = -20 °C) |
|---|---|---|---|
| [NaMg (Sea salt)] | -0.162 | 0.127 | -0.067 |
| [AlSiCa (Dust)] | 0.147 | 0.095 | **0.228** |
| [P (Nutrient)] | -0.166 | -0.072 | **0.233** |
| [S (DMS)] | NA | 0.366$^*$ | 0.435$^*$ |

**Table B3.** Pearson correlation coefficients ($r$) calculated between $N_{INP}$ at selected freezing temperatures and the percentage of time that 2-day backward trajectories have spent over the ocean, land, MIZ, and ice pack. $r$ values in bold represent results with statistical significance ($p < 0.05$). $r$ values with $*$ denote moderate correlations ($0.3 < |r| < 0.7$), and with $**$ indicate strong correlations ($|r| > 0.7$).

| Surface type | $N_{INP}$ (T = -15 °C) | $N_{INP}$ (T = -20 °C) | $N_{INP}$ (T = -30 °C) | $N_{INP}$ (T = -34 °C) |
|---|---|---|---|---|
| Ocean | 0.143 | -0.112 | 0.233 | 0.153 |
| Land | -0.047 | 0.016 | 0.219 | 0.066 |
| MIZ | -0.155 | 0.089 | 0.296 | -0.019 |
| Ice-pack | -0.149 | -0.165 | 0.284 | -0.018 |

**Appendix C: Time series of $N_{INP}$ and the concentration of elemental tracers for dust, sea salt, nutrients, and DMS.**





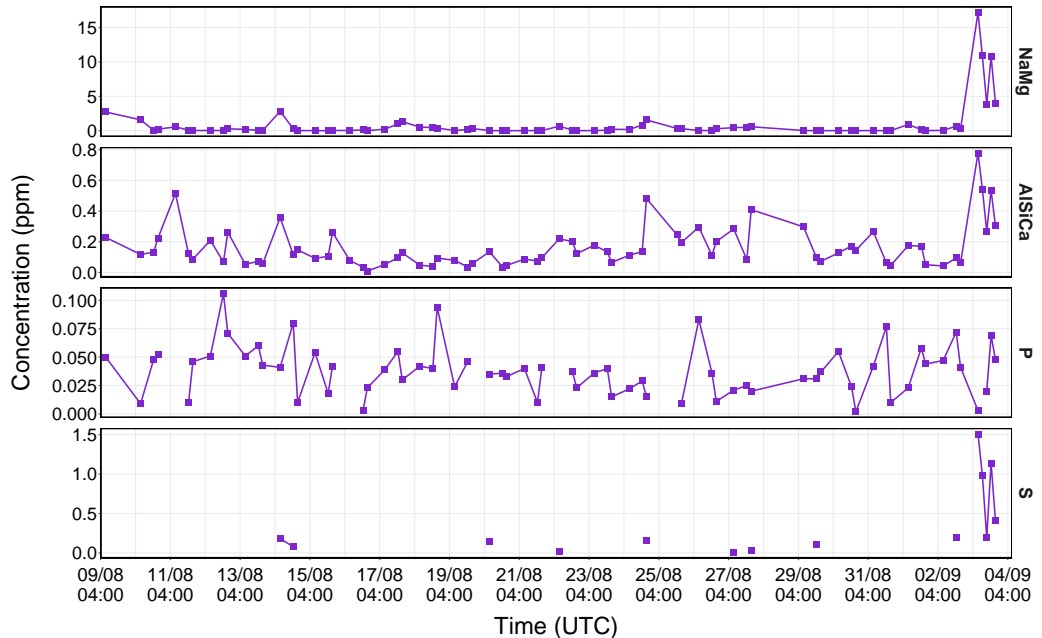

**Figure C1.** Time series of concentrations of selected elemental species (from top to bottom panels: Na and Mg (sea salt), AlSiCa (dust), P and S) measured from impinger samples using ICP-OES. Only results above the LOD are shown.

## Appendix D:  Geographical variability of $N_{\mathrm{INP}}$ along the ship track measured from impinger samples



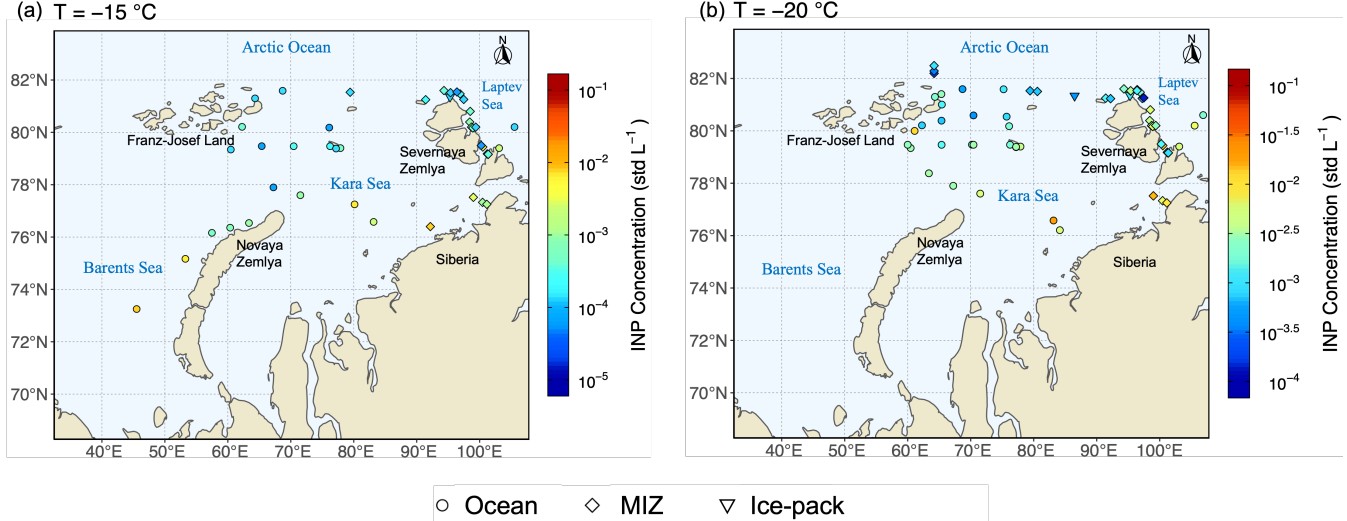

**Figure D1.** Geographical variability of INP concentrations during the Arctic Century Expedition at (a) *T* = -15 °C and (b) *T* = -20 °C measured with impinger samples and DRINCZ. Different shapes represent the categories of sea ice extent when the measurements were taken based on the daily observation of sea ice concentrations.

*Data availability.* The data presented in this study will be made available at https://doi.org/10.3929/ethz-b-000717424 after acceptance of
the publication.

*Author contributions.* GL performed sample processing and data analysis, produced figures, interpreted results, and wrote the original manuscript draft. GL and AW participated in the campaign and conducted in situ sampling and measurements. IT provided the LAGRANTO backward trajectory data. AW, IT, and UL provided feedback on data interpretation. ZAK supervised the project, obtained funding, and was involved in experiment planning, data interpretation, and manuscript writing. All authors reviewed the manuscript.

*Competing interests.* The authors declare that no competing interests are present.

*Acknowledgements.* This research used samples and/or data provided by the Arctic Century Expedition, a joint initiative led by the Swiss Polar Institute (SPI), the Antarctic and Arctic Research Institute (AARI) and GEOMAR Helmholtz Centre for Ocean Research Kiel (GEOMAR) and funded by the Swiss Polar Foundation. This project has received funding from the Horizon Europe Program under Grant Agreement NO. 101137680 via project CERTAINTY (Cloud aERosol inTeractions & their impActs IN The earth sYstem). GL and ZAK acknowledge that
this project has been made possible by a grant from the Swiss Polar Institute, Dr. Frederik Paulsen. We acknowledge all those involved in the fieldwork associated with the Arctic Century Expedition, including technical support from Dr. Michael Rösch. We would like to thank Dr.





Xu Fang from ETH for providing the ICP-OES instrument, with assistance on sample preparation and measurements. We thank Franziska Aemisegger for the calculation of the backwards trajectories.



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
