# Peer review of "Ice Nucleating Particle Concentrations over the Eurasian-Arctic seas"

_EGUsphere, 2025_

## Referee Comment (RC2)

**Review of "Ice Nucleating Particle Concentrations Over the Eurasian-Arctic Sea"**

Authors: Guangyu Li, André Welti, Iris Thurnherr, Ulrike Lohmann, and Zamin A. Kanji

This paper investigates ice nucleating particle concentration (nINP) in the Eurasian Arctic seas during the summer of 2021. The authors collected data from aboard the RV Akademik Tryoshnikov and deployed both DRINCZ (offline) and HELC (online) measurement techniques to collect and assess INP activity. The researchers utilized the LAGRANTO tool for backward air parcel trajectories, categorizing them by passed-over surface types, including ice-pack, the ice-free ocean, and land. Additional testing was performed to find chemical composition and particle size distribution. The research revealed the highest nINP near Novaya Zemlya, and linked the air parcels to originating on the Siberian coast. While the authors found evidence of high terrestrial impact, statistical tests revealed negligible correlation between nINP and air-parcel time over land, suggesting local dominance of INPs over long-range transport. The manuscript is well organized and easy to follow with a strong experimental design. The reviewers recommend publication after revision of the following minor comments.

**Minor Comments:**

**Title:**

- By adding "During Summer" to the end of the title, the author can better articulate what the manuscript has to offer.

**Page 4, Section 2.2.1 Ambient aerosol samples:** *Lines 80-81*

- How were subsequent flow rates chosen for high flow rate and low volume samplers? The author mentions that the high volume of air being processed in the impinger samples affects the detectable range of nINP, but does not justify why that flow rate was selected.
- Could the flow rates be optimized?

Page 4, Section 2.3.1 Impinger and filter samples: Line 89

- The author uses the acronym DRINCZ without defining. This reviewer thinks it would be better to introduce the instrument by its full name. (Especially since the next section is about it).

**Page 6, Section 2.4.2 Chemical Composition Analysis:** *Line 133; Line 135*

- What is the accuracy and uncertainty of the bulk instrument (ICP-OES) technique? Was it dependent on the volume of the sample to be analyzed? Dependent on the element?
- Why were ICP-OES samples diluted with Nitric acid? Was the instrument calibrated using standards in HNO3, etc., or was it to ionize metals?

**Page 7, Section 2.4.4 Backwards Trajectory Analysis:** *Line 160;*

- The authors are presuming that >0.1mm/h of rain was chosen due to its potential to wash out aerosols from the airmass. If so, it should be stated in the manuscript for clarity.
- Why were the trajectories calculated for two days (48hrs)? Why not longer? Does the shorter time for back trajectory limit the number of long-range aerosols being observed?

**Page 7, Section 3.1 Temperature-dependent variability of Arctic INP Concentrations:** *Lines 174-175;*

- Isn't LOD a part of the reason for the observed "smaller" difference? The line at 5x10^-5 INP L^-1 above -17 dC is virtually horizontal. Has this been considered? If so it should be stated in the manuscript, otherwise the author should justify.
- To this reviewer, the variability and difference look larger and more prominent at higher freezing temperatures.

**Page 8, Section 3.1 Temperature-dependent variability of Arctic INP Concentrations:** *Lines 185-186; Figure 2*

- What makes the author say this? To the reviewer, nINP does not look similar or overlapping in the min-max range, even at -15 dC. The authors' measurements represent lower nINP overall.
- Are the references used all coming from summertime? Anything including the data from the Arctic spring, which may not be directly comparable?
- It's better to compare the freezing temperature-binned averages and/or medians from each study.
- Figure 2 is difficult to read. The number of references seems excessive for no reason. What is the justification for each reference, and why are they included here?

**Pages 9-10, Section 3.1 Temperature-dependent variability of Arctic IINP Concentrations:**
*Line 195, Line 196, Figures 3 and 4*

- 1 in 10^5 particles seems pretty high as compared to previous measured arctic INP fractions from Spitsbergen reported in Rhinaldi et al. (2021) (ACP at -18 dC) and from Alaskan Arctic (Pantoya et al, (2025) AR at -20 dC). Their previous values are an order of magnitude lower at even lower freezing temperatures. The author should discuss why.
- Are there any other reports on freezing fraction or IN efficiency from other arctic areas for comparison?
- Shouldn't the correlation be looked into for similar local meteorological conditions? Could be dependent on wind speed and direction, etc.
- In figures 3 and 4, the legend should be put at the top (like in figure 8). It is clearer.

**Page 11, Section 3.3 Chemical Composition and Sources of INPs over the Eurasian-Arctic Ocean:** *Line 204, Line 214;*

- The reviewer believes it should be made more clear why the author chose to use impinger samples. Impinger collects large particles and can miss some fine particles. Wouldn't this bias the results?
- From Fig. 2 it looks like impinger data show lower nINP as compared to the data from the other assay and previous studies.
- Suggesting phytoplankton blooms can enrich the INP population is speculative without evidence.
- What season is the bloom in the author's study area? In some locations there is a time lag between phytoplankton bloom and the release of DMS and other byproducts.

**Page 12, Section 3.3 Chemical Composition and Sources of INPs over the Eurasian-Arctic Ocean:** *Figure 4*

- How did the authors estimate PPM of each categorized species? Function of total aerosol concentration and dilution factor during sample prep? Please clarify it in section 2.4.2

**Page 13, Section 3.4 Geographical variability of INP concentrations across the Arctic marine and ice landscape:** *Lines 27-28*

- Other more relevant papers seem better suited here. The reviewer suggests replacing the current citation with something like:
  (https://iopscience.iop.org/article/10.1088/1748-9326/ab87d3).

**Page 14-15, Section 3.4 Geographical variability of INP concentration across the Arctic marine and ice landscape:** *Figures 5, 6*

- Was there any impact of local precipitation and wind properties on nINP or ice active fraction?
- The reviewer thinks that the better matrix to present additionally with Figure 6 is the activated fraction spectra (or other spectra showing IN efficiency).
- The authors have PSD data alongside their INP data. Does the nucleation efficiency show the same trend as nINP, and give the same conclusion? If not, it should be explained why.

**Page 16-18, Section 3.5 Influence of Air mass origins on INP variability:** *Lines 236-237, Figure 7*

- Between local terrestrial sources and long-range aerosol transport, which is more important? Why?
- Is the air mass traveling near the ground surface or aloft? No information is provided on air mass altitude.
- Figure 7 or Table B3 do not offer the site specific altitude information.
- Perhaps the author could offer the time series of altitudes as subpanels in Fig. 9 for high- and low- INP episodes.
- Figure 7 is very busy. It is difficult to find a takeaway message from this figure .

**Pages 19-21, Section 3.6 Case Study:** *Figure 9, 10, and Line 284*

- The author should consider mentioning somewhere in this section the similarities in Pantoya et at. (2025) where they showed low INP episodes from the north and high arctic from increased amounts of ice pack contribution, and high INP episodes from south/mid-latitude, both observed in the Alaskan Arctic. This provides an example of a longer sampling period.
  (https://ar.copernicus.org/articles/3/253/2025/ar-3-253-2025.html)
- Figure 9 could be improved with the addition of air mass altitude information.
- In line 284 the author uses a citation that should include:
  https://doi.org/10.1029/2021GL094646
- Figure 10 should include an IN efficiency comparison figure. Does nINP scale to aerosol concentration? This may provide insight for the author on INP composition and source.

- What about the role of organic particles? Is there any indication that biomass burning can be a source of high INPs?
- Any large/local biomass burning events coincide with the authors measurement periods?

**Page 22, Section 4 Summary and Conclusions:** *Lines 312-113, Lines 315-316*

- In lines 312-313, perhaps the author should discuss what we expect to see in different seasons, especially in the arctic spring, when arctic haze often prevails and delivers mid-lat emissions to the arctic.
- In line 315, the author states that "future research" is needed. What research is needed? The reviewer suggests that the community needs longer INP monitoring and vertical distributions. This provides a more concrete outlook for researchers and reviewers.

**Page 27, Appendix D**: *Figure D1*

- Why doesn't MIZ near Sevemaya Zemlya show high INPs? This seems inconsistent with the previous figure (Figure 5).

**Minor Edits:**

- Page 2, Line 26: "origins of INP" should be "origins of INPs".
- Page 2, Line 31: "spend" should be changed to "spent".
- Page 2, Line 40: "aerosol" should be changed to "aerosols".
- Page 2, Line 51: "MBA emission" should be "MBA emissions".
- Page 4, Line 79: A space should be added between "10um".
- Page 4, Line 86: Change "fridge" to "refrigerator".
- Page 5, Line 101: The author uses "nano-pure water" here instead of "ultra-pure water" which is used in the method section. Consistency will help eliminate any confusion.
- Page 6, Line 130: "sphere" should be changed to "spherical".
- Page 6, Line 145: "directions" should be "direction".
- Page 7, Line 154: "Backwards" should be "Backward" when referring to trajectory (in figures as well).
- Page 7, Line 168: The word "for" should be inserted after "except".
- Page 8, Line 182: "see (Tobo et al., 2019)" should be "see Tobo et al., (2019)".

- Page 19, Line 270: The typo "occured" should be corrected to "occurred".
- The author should remain consistent when referring to plural/non-plural parameters. Many times in the manuscript the author switches mid-sentence. (e.g. Page 24, Line 333: "from *filter* compared to impinger samples" should be "from *filters* compared to impinger samples").